# An Outbreak of ST859-K19 Carbapenem-Resistant Hypervirulent *Klebsiella pneumoniae* in a Chinese Teaching Hospital

Junying Zhu,[a] Xuemei Jiang,[b] Lina Zhao,[a] Min Li[a]

aDepartment of Laboratory Medicine, Ren Ji Hospital, School of Medicine, Shanghai Jiao Tong University, Shanghai, China
bClinical Laboratory, Urumqi Maternal and Child Health Care Hospital, Urumqi, Xinjiang, China

Junying Zhu and Xuemei Jiang contributed equally to this work. Author order was determined alphabetically.

**ABSTRACT** Carbapenem-resistant hypervirulent *Klebsiella pneumoniae* (CR-hvKP) has been increasingly reported worldwide. Here, we report an outbreak caused by sequence type 859-K19 (ST859-K19) CR-hvKP isolates in a teaching hospital in China. Interestingly, *K. pneumoniae* ST859 was a single-locus variant of ST11 but has never been reported before. A total of 11 nonrepetitive ST859 CR-hvKP isolates were collected from 11 patients, 3 of which died of severe CR-hvKP infection. Antimicrobial susceptibility assay results showed that all the 11 CR-hvKP isolates exhibited high-level resistance to commonly used antibiotics, only remaining susceptible to colistin, tigecycline, and ceftazidime/avibactam. Whole-genome sequencing (WGS) was performed using the Illumina platform for the 11 CR-hvKP isolates, and RJ9299 was further sequenced using the PacBio platform. A phylogram tree using WGS data revealed that all the 11 CR-hvKP isolates were clustered in 1 clade, which probably indicated clone transmission. Determinants of resistance and virulence gene analysis using WGS data confirmed the 11 isolates had almost identical resistance gene profiles ($bla_{KPC-2}$, $bla_{TEM-1B}$, $bla_{SHV-187}$, $rmtB$, $fosA6$) and virulence gene ($rmpA$, $rmpA2$, $iucABCDiutA$) profiles, which hint at clone spread again. The complete genome size of RJ9299 was 5,875 kb, including a 5,445-kb chromosome, a 215-kb virulence plasmid (pVir-CR-hvKP-RJ9299), a 109-kb $bla_{KPC-2}$-harboring plasmid (pKPC-2-RJ9299), and three circular plasmids. Comparative genomics showed pVir-RJ9299 (IncHI1B type) and pKPC-2-RJ9299 (IncFII-IncR) possessed over 99% similarity to pLVPK and pKPC-CR-hvKP-C789, respectively. Serum resistance assays and a *Galleria mellonella* infection model showed the 11 isolates exhibited different levels of virulence. This is the first report of an outbreak caused by ST859 CR-hvKP isolates.

**IMPORTANCE** The emergence of carbapenem-resistant hypervirulent *Klebsiella pneumoniae* (CR-hvKP) in China has posed a great threat to public health, especially in the highly transmissible ST11 clone. With the transmission of virulence and resistance determinants, CR-hvKP isolates have been reported in an increasing number of sequence types (STs), including ST23, ST65, ST1797, ST43, ST231, ST147, ST15, ST383, ST268, ST595, ST375, ST48, and ST307. Here, we report an outbreak caused by ST859-K19 CR-hvKP isolates in a teaching hospital in China. ST859 is a single-locus variant of ST11. There is no literature on ST859 *K. pneumoniae* in public databases, let alone ST859 CR-hvKP isolates. To our knowledge, this is the first report to depict the molecular and genomic characteristics of ST859 CR-hvKP isolates. Active surveillance approaches should be implemented to promptly find the spread of CR-hvKP isolates in health care settings.

**KEYWORDS** outbreak, carbapenem-resistant, hypervirulent, *Klebsiella pneumoniae*, KPC-2

Address correspondence to Min Li, ruth_limin@126.com.

The authors declare no conflict of interest.

**K**lebsiella pneumoniae, familiar to clinicians, is capable of causing pneumonia, bacteremia, pyogenic liver abscess, and urinary tract infections (1). In recent years, two major types of clinically significant pathogens, namely, carbapenem-resistant *K. pneumoniae* (CRKP)

and hypervirulent *K. pneumoniae* (hvKP), have driven a number of studies and attracted extensive attention worldwide (2). Horizontal transfer of mobile genetic elements (MGE) promoted the spread and transmission of CRKP. According to China Antimicrobial Surveillance Network (CHINET), the prevalence of CRKP (defined by resistance to any of the carbapenems) increased from 3.0% in 2005 to 20.9% in 2017 in China (3). Infections caused by CRKP mainly occurred in health care settings and have high rates of morbidity and mortality due to the scarcity of effective treatments (4). In China, Klebsiella pneumoniae carbapenemase-2 (KPC-2) was the main genetic determinant of CRKP, accounting for approximately 70%, followed by New Delhi Metallo-beta-lactamase-1 (NDM-1) and OXA-48 (5). A number of sequence types (STs) in CRKP have been reported globally. The ST11 clone is predominantly found in China and South America, ST258 is mostly reported in the United States, ST512 is endemic in Italy and Greece, and ST147 is mainly reported in India and Tunisia (6–9).

On the other hand, hvKP frequently causes community-acquired, severe, acute, and deadly pyogenic liver abscesses in young and healthy individuals (10). Yu tang et al. (11) evaluated the in-hospital mortality of patients with hvKP infections and found that 11.9% of patients died of hvKP infections. In general, hvKP is mainly restricted to the ST23 and ST65 clonal background. Compared with classic *Klebsiella pneumoniae* (cKP), hvKP often showed substantially higher virulence, which was attributed to the carriage of virulence plasmids (e.g., pLVPK). These plasmids encode a variety of virulence factors, such as the aerobactin synthesis operon (*iucABCD*), the outer membrane ferric aerobactin receptor (*iutA*), a gene cluster of salmochelin production (*iroBCDN*), and regulator of mucoid phenotype A (*rmpA* and/or *rmpA2*) (12).

However, with the transmission and evolution of plasmids, another "superbug," namely, CR-hvKP, in which carbapenem resistance and hypervirulence in highly transmissible epidemic clones converged, has emerged as a global crisis. Gu et al. (13) reported a fatal outbreak caused by ST11 CR-hvKP, which is simultaneously hypervirulent, multidrug-resistant, and highly transmissible, in a Chinese hospital. In addition, CR-hvKP was also reported in other STs, including ST23, ST65, ST1797, ST43, ST231, ST147, ST15, ST383, ST268, ST595, ST375, ST48, and ST307 (14–19). In this study, we report an outbreak caused by a KPC-2-producing ST859 CR-hvKP clone which is a new ST and single-locus variant of ST11.

## RESULTS

**Clinical characteristics of 11 CR-hvKP isolates.** The 11 CR-hvKP isolates were isolated from 11 patients (8 patients from the intensive care unit [ICU], 2 patients from neurosurgery, and 1 general surgery inpatient) between October 2019 and March 2020. The median age of the 11 patients (6 males and 5 females) was 62 years (range, 36 to 74 years). The patients had various underlying diseases, including hypertension, cerebral hemorrhage and tumor. In addition, two patients had suffered severe craniocerebral injury. The length of hospital stays of the 11 patients varied from 10 days to 90 days. All the patients received antimicrobial treatment, including carbapenem alone or in combination with other alternative antibiotics when necessary. In the end, seven patients developed pneumonia, and three of those patients died of CR-hvKP infection; the other four patients were discharged. Nine CR-hvKP isolates were collected from sputum, and another two isolates were, respectively, from blood and drainage (Table 1 and Fig. 1).

**Multilocus sequence typing and antimicrobial susceptibility testing.** Multilocus sequence typing (MLST) revealed that all the 11 isolates are exclusively grouped into ST859, which is a single-locus (*tonB*) variant of ST11, with the *tonB1* gene in ST859 differing from *tonB4* in ST11. Antimicrobial susceptibility assay results showed that the 11 isolates had almost identical antibacterial susceptibility profiles. All the isolates exhibited high-level resistance to imipenem (MIC range, 32 to 128 $\mu$g/mL), meropenem (MIC range, 128 to 256 $\mu$g/mL), ceftazidime (MIC, 128 $\mu$g/mL), cefepime (MIC, 128 $\mu$g/mL), aztreonam (MIC, 128 $\mu$g/mL), cefoperazone/sulbactam (MIC, 128 $\mu$g/mL), and levofloxacin (MIC range, 8 to 32 $\mu$g/mL) while remaining susceptible to colistin (MIC range, 0.125 to 0.5 $\mu$g/mL), tigecycline (MIC range, 0.125 to 0.25 $\mu$g/mL), and ceftazidime/avibactam (MIC range, 2 to 4 $\mu$g/mL). Moreover, 10 (83.3%) isolates showed resistance to amikacin (MIC, 128 $\mu$g/mL) and gentamicin (MIC, 128 $\mu$g/mL) (Table 2).

**TABLE 1** Clinical history of 11 patients carrying ST859 CR-hvKP isolates[a]

| Isolate | Age (yrs)/sex | Isolation date (mo/day/yr) | Specimen | Ward | Hospital days | Antibiotic treatment | Underlying disease(s) | Outcome |
|---|---|---|---|---|---|---|---|---|
| RJ 9299 | 62/M | 10/24/2019 | Sputum | ICU | 90 | CXM, FEP, MEM, LEV, SCF, COL, IPM, TGC, TZP | Cerebral infarction | Discharged |
| RJ 9337 | 67/F | 11/2/2019 | Drainage | General surgery | 31 | CXM, CRO, IPM, MEM, FEP | Stomach cancer | Improvement |
| RJ 9490 | 74/M | 11/21/2019 | Sputum | ICU | 90 | SCF, FEP, MXF, CRO, AMK, MEM, ATM, LEV, CAZ | Pneumonia, cerebral hemorrhage, hypertension | Improvement |
| RJ 9582 | 52/M | 12/5/2019 | Sputum | ICU | 10 | CXM, KZ, FOX, IPM, TGC, SCF | Hypertensive cerebral hemorrhage, pneumonia, septicopyemia | Discharged |
| RJ 9690 | 69/M | 12/20/2019 | Blood | ICU | 11 | FEP, SCF, MEM | Cerebral hemorrhage, pneumonia | Dead |
| RJ 9717 | 57/F | 12/23/2019 | Sputum | ICU | 49 | KZ, FOX, FEP, MXF, SCF, CRO, MEM, AMK, COL | Severe craniocerebral injury, rib fracture | Discharged |
| RJ 9752 | 68/M | 12/30/2019 | Sputum | ICU | 11 | CXM, CRO, FEP, CAZ, MEM | Diabetes, hypertension, hyperuricemia | Dead |
| RJ 9846 | 51/F | 1/15/2020 | Sputum | ICU | 13 | KZ, FOX, MEM | Intracranial mass lesion | Discharged |
| RJ 9860 | 36/M | 1/11/2020 | Sputum | Neurosurgery | 51 | KZ, VAN, MEM, IPM, SCF, CZA | Severe craniocerebral injury | Discharged |
| RJ 9950 | 55/F | 1/28/2020 | Sputum | ICU | 36 | KZ, VAN, MEM, LEV, FEP, CZA | Hypertension, cerebral, hemorrhage, pneumonia | Dead |
| RJ 10129 | 64/M | 3/9/2020 | Sputum | Neurosurgery | 33 | KZ, VAN, CRO, SCF, MEM | Hypertension, cerebral hemorrhage | Discharged |

[a]M, male; F, female; ICU, intensive care unit; CXM, cefuroxime; FEP, cefepime; MEM, meropenem; LEV, levofloxacin; SCF, cefperazone sulbactam; COL, colistin; IPM, imipenem; TGC, tigecycline; TZP, piperacillin tazobactam; CRO, ceftriaxone; MXF, moxifloxacin; AMK, amikacin; ATM, aztreonam; CAZ, ceftazidime; KZ, cefazolin; FOX, cefoxitin; TGC, tigecycline; VAN, vancomycin; CZA, ceftazidime/avibactam.

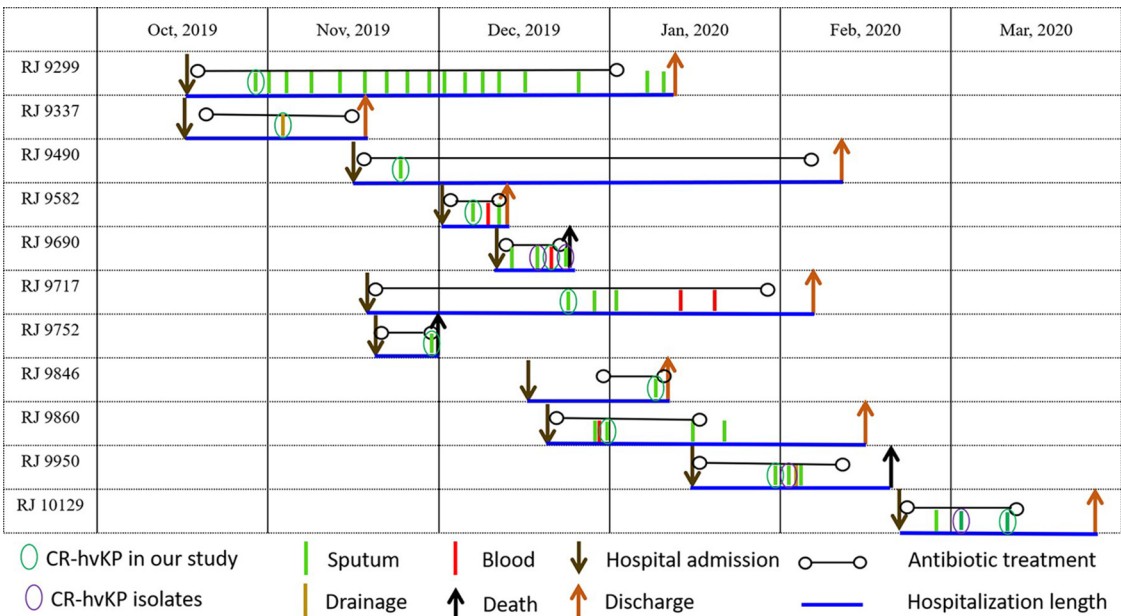

**FIG 1** Epidemiology of the carbapenem-resistant hypervirulent *K. pneumoniae* outbreak cases.

**Resistance gene, virulence gene profiles, and phylogenetic analysis.** Genome data also showed that these 11 isolates belonged to ST859 and the K19 locus. Interestingly, the 11 isolates also showed identical resistance and virulence gene profiles. All the 11 isolates carried $bla_{KPC-2}$, $bla_{TEM-1}$, and *fosA*. Seven isolates harbored both *rmtB* and *qnrS* genes, two isolates only carried *rmtB*, and another two isolates only carried *qnrS*. Except for RJ9717, which carried $bla_{SHV-12}$, all the other isolates carried $bla_{SHV-187}$. In terms of virulence genes, they all carried regulator of mucoid phenotype A (*rmpA* and *rmpA2*), aerobactin (*iucABCDiutA*), salmochelin (*iroE*), type 3 fimbriae (*mrkABCDF*) and type 1 fimbrial adhesion genes (*fimA-H*), enterobactin (*entABCDEFS*, *fepABCDG*, *ybdA*, and *fes*), and yersiniabactin (*ybtAEPQSTUX*, *irp1/2*, and *fyuA*) (Table 3).

We mainly focused on the ST859 outbreak isolates but included 15 genomes of ST11 *K. pneumoniae* isolates to generate a phylogenetic tree (Fig. 2A). A total of 5,901 single-nucleotide polymorphisms (SNPs) between the 11 ST859 outbreak isolates and 15 genomes of ST11 *K. pneumoniae* were identified and used to construct the maximum likelihood tree. The population framework revealed a diverse population structure containing three distinct clades (clade I, clade II, clade III). The 11 ST859-K19 CR-hvKP isolates were clustered

**TABLE 2** String test, K-locus, MLST, and antimicrobial susceptibility results of 11 CR-hvKP isolates[a]

| Isolate | String test | K | ST | MIC (µg/mL) for: | | | | | | | | | | | |
|---|---|---|---|---|---|---|---|---|---|---|---|---|---|---|---|
| | | | | AMK | GEN | LEV | CZA | FEP | SCF | COL | CAZ | IPM | MEM | ATM | TGC |
| RJ 9299 | + | K19 | ST859 | 128 | 128 | 32 | 2 | 128 | 128 | 0.125 | 128 | 128 | 256 | 128 | 0.25 |
| RJ 9337 | + | K19 | ST859 | 128 | 128 | 8 | 2 | 128 | 128 | 0.125 | 128 | 64 | 256 | 128 | 0.25 |
| RJ 9490 | + | K19 | ST859 | 128 | 128 | 8 | 2 | 128 | 128 | 0.125 | 128 | 64 | 256 | 128 | 0.25 |
| RJ 9582 | + | K19 | ST859 | 128 | 128 | 8 | 2 | 128 | 128 | 0.125 | 128 | 64 | 256 | 128 | 0.125 |
| RJ 9690 | + | K19 | ST859 | 128 | 128 | 8 | 2 | 128 | 128 | 0.125 | 128 | 64 | 256 | 128 | 0.125 |
| RJ 9717 | + | K19 | ST859 | 128 | 128 | 8 | 2 | 128 | 128 | 0.125 | 128 | 64 | 128 | 128 | 0.25 |
| RJ 9752 | + | K19 | ST859 | 128 | 128 | 8 | 2 | 128 | 128 | 0.25 | 128 | 64 | 256 | 128 | 0.125 |
| RJ 9846 | + | K19 | ST859 | 2 | 0.5 | 8 | 2 | 128 | 128 | 0.125 | 128 | 64 | 256 | 128 | 0.125 |
| RJ 9860 | + | K19 | ST859 | 128 | 128 | 16 | 4 | 128 | 128 | 0.5 | 128 | 32 | 256 | 128 | 0.25 |
| RJ 9950 | + | K19 | ST859 | 128 | 128 | 16 | 2 | 128 | 128 | 0.25 | 128 | 64 | 256 | 128 | 0.125 |
| RJ 10129 | + | K19 | ST859 | 128 | 128 | 8 | 2 | 128 | 128 | 0.25 | 128 | 64 | 256 | 128 | 0.125 |
| ATCC 25922 | NA | NA | NA | 0.5 | 1 | 0.06 | 0.125 | 0.06 | 0.06 | 0.25 | 0.25 | 0.125 | 0.125 | 0.125 | 0.125 |

[a]+, positive; NA, not applicable; AMK, amikacin; GEN, gentamicin; LEV, levofloxacin; CZA, ceftazidime-avibactam; FEP, cefepime; SCF, cefoperazone/sulbactam; COL, colistin; CAZ, ceftazidime; IPM, imipenem; MEM, meropenem; ATM, aztreonam; TGC, tigecycline.

**TABLE 3** Resistance gene and virulence gene profiles of 11 CR-hvKP isolate[a]

| | Resistance genes | | | | | | | | | Virulence genes | | | | | | | |
|---|---|---|---|---|---|---|---|---|---|---|---|---|---|---|---|---|---|
| | Carbapenemases | | | ESBLs | | | 16S RMTase genes | PMQR genes | fosA6 | | | | | | | | |
| Isolate | KPC-2 | NDM | OXA | CTX-M | SHV[-] | TEM-1B | rmtB | qnrS1 | fosA6 | rmpA | rmpA2 | iucABCD, iutA | iroE | mrkABCDF | fimA-H | entABCDEFS, fepABCDG, ybdAi, fes | ybtAEPQSTUX, irp1/2, fyuA |
| RJ9299 | + | − | − | − | 187 | + | + | + | + | + | + | + | + | + | + | + | + |
| RJ 9337 | + | − | − | − | 187 | + | + | − | + | + | + | + | + | + | + | + | + |
| RJ 9490 | + | − | − | − | 187 | + | + | − | + | + | + | + | + | + | + | + | + |
| RJ 9582 | + | − | − | − | 187 | + | + | + | + | + | + | + | + | + | + | + | + |
| RJ 9690 | + | − | − | − | 187 | + | + | + | + | + | + | + | + | + | + | + | + |
| RJ 9717 | + | − | − | − | 12 | + | + | + | + | + | + | + | + | + | + | + | + |
| RJ 9752 | + | − | − | − | 187 | + | − | + | + | + | + | + | + | + | + | + | + |
| RJ 9846 | + | − | − | − | 187 | + | + | + | + | + | + | + | + | + | + | + | + |
| RJ 9860 | + | − | − | − | 187 | + | + | + | + | + | + | + | + | + | + | + | + |
| RJ 9950 | + | − | − | − | 187 | + | + | + | + | + | + | + | + | + | + | + | + |
| RJ 10129 | + | − | − | − | 187 | + | − | + | + | + | + | + | + | + | + | + | + |

[a]+, positive; −, negative; 16S-RMTase, 16S ribosomal RNA methyltransferase; PMQR, Plasmid-mediated quinolone resistance; SHV-, positive for SHV variants.

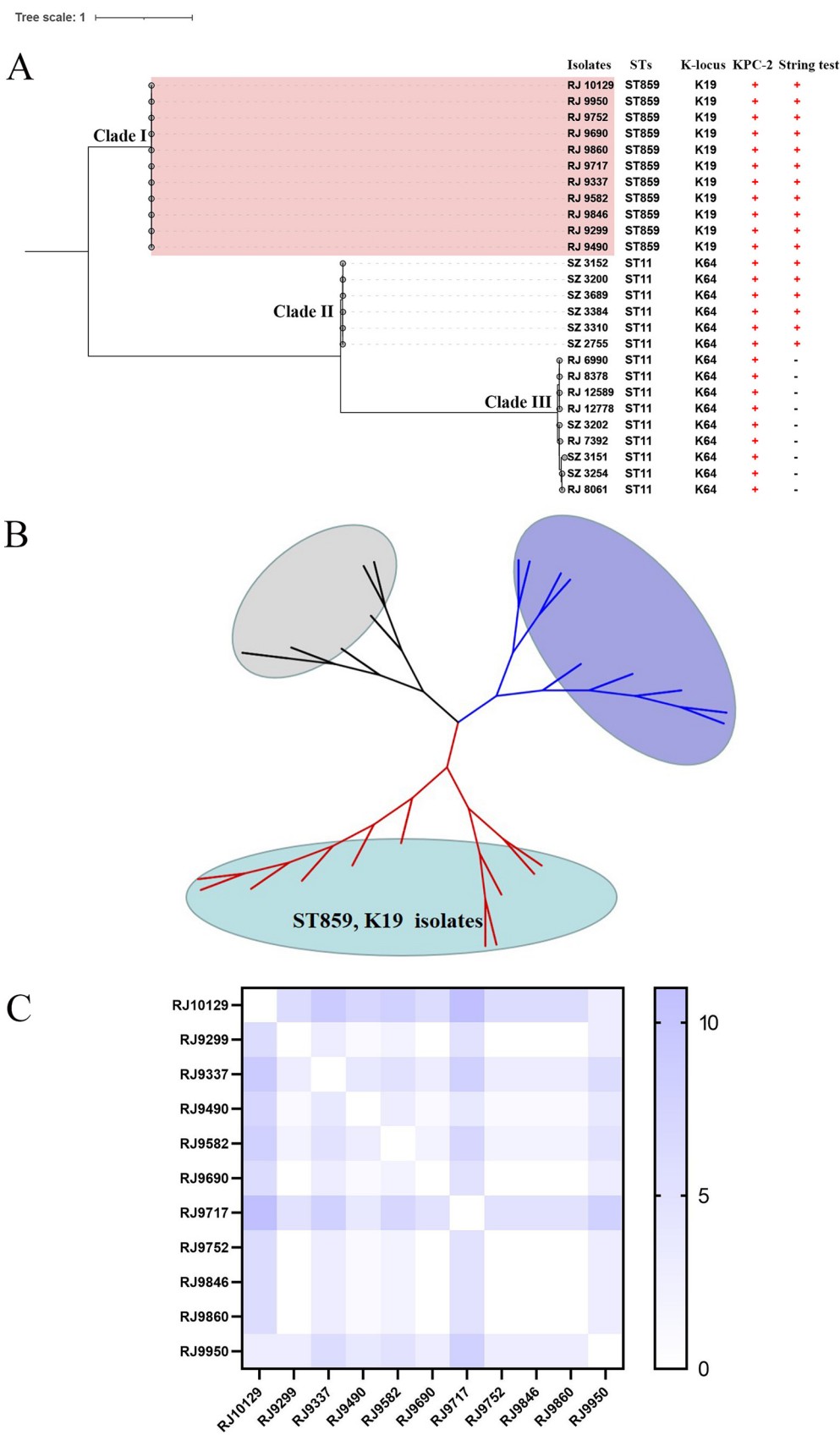

**FIG 2** (A) Phylogenetic structure and paired SNP distance of *K. pneumoniae* maximum likelihood tree of ST859 and ST11 clonotype *K. pneumoniae* isolates. (B) Unrooted tree of ST11 and ST859 clonotype *K. pneumoniae* isolates. Branches could be classified into three clades. (C) Paired SNP distance of ST859 clonotype *K. pneumoniae* isolates.

into clade I. Six ST11-K64 KPC-2-producing isolates which had a positive string test, namely, CR-hvKP, were clustered into clade II. Nine ST11-K64 KPC-2-producing isolates which had a negative string test, namely, CRKP, were clustered into clade III. Furthermore, the three clades could be markedly distinguished in the unrooted tree (Fig. 2B). Paired distance between the 11 ST859 isolates was used to define the relatedness of the isolates. As shown in Fig. 2C, our analysis revealed 15 SNPs at most (minimum, 0) when comparing 10 CR-hvKP isolates against the reference genome of RJ9299, which indicated probable clonal transmission.

**Genome characteristics of RJ9299.** We selected RJ9299 for further sequencing with the PacBio platform. The genome size was 5,875,454 bp, including a chromosome of 5,445,578 bp and five circular plasmids, a virulence plasmid (pVir-CR-hvKP-RJ9299), a $bla_{KPC-2}$-harboring plasmid (pKPC-2-RJ9299), plasmidC-RJ9299, plasmidD-RJ9299, and plasmidE-RJ9299. Using the assembled contigs to query MLST and the wzi allele (https://bigsdb.pasteur.fr/cgi-bin/bigsdb/bigsdb.pl?db=pubmlst_klebsiella_seqdef&page=sequenceQuery) databases also showed that RJ9299 belonged to ST859 and the K19 serotype (wzi_allele 19).

Virulence gene analysis revealed that a set of virulence factors was located on the chromosome, including type 3 fimbriae (*mrkABCDF*) and type 1 fimbrial adhesion genes (*fimA-H*), enterobactin (*entABCDEFS*, *fepABCDG*, *ybdA*, and *fes*), salmochelin (*iroE*), and yersiniabactin (*ybtAEPQSTUX*, *irp1/2*, and *fyuA*). Apart from virulence genes on the chromosome, the chromosomal integration of the integrative and conjugative element ICE*Kp5* (*ybt14*) was spotted in a specific asparagine-tRNA gene.

Comparative genomics showed that pVir-CR-hvKP-RJ9299 possessed over 99% similarity (with 92% query coverage) to pLVPK (GenBank accession no. AY378100) and 99% similarity (with 81% query coverage) to pVir-CR-HvKP4 (MF437313) (Fig. 3). Similar to the pLVPK plasmid (size, 219,385 bp), which belonged to the IncHI1B-IncFIB replicon incompatibility type, pVir-CR-hvKP-RJ9299 was an IncHI1B-type plasmid with a length of 215,959 bp and an average GC content of 49.87%. Virulence gene analysis indicated that pVir-CR-hvKP-RJ9299 carried a mass of virulence genes, including mucoid regulator-encoding genes (*rmpA*, *rmpA2*) and aerobactin (*iucABCD* and *iutA*).

However, unlike previous versions of ~170-kb KPC-2-encoding plasmids, the size of pKPC-2-RJ9299, which belonged to the IncFII-IncR type, was ~109 kb. Sequence alignment showed pKPC-2-RJ9299 possess over 99% similarity (with 99% query coverage) to pKPC-L388 (CP029225.1) and 100% similarity (with 99% query coverage) to pKPC-CR-hvKP-C789 (CP034417.1) (Fig. 4A). Importantly, sequence analysis indicated that pKPC-2-RJ9299 also carried additional antibiotic resistance determinants, including extended-spectrum $\beta$-lactamase genes ($bla_{TEM-1}$, $bla_{SHV-187}$) and an aminoglycoside resistance gene (*rmtB*). Various mobile elements, such as Tn*2* and IS*Kpn6*, were located upstream and downstream of the aforementioned antimicrobial resistance genes (Fig. 4B). Linear genomic alignment revealed that an almost exactly identical cassette (IS*Kpn6*-$bla_{KPC-2}$-IS*Kpn27*-IS26) was surrounding $bla_{KPC-2}$ between pKPC-CR-hvKP-C789 and pKPC-2-RJ9299.

The other three small plasmids, plasmid C-RJ9299 (87,096 kb), plasmid D-RJ9299 (11,970 kb), and plasmid E-RJ9299 (5,596 kb) did not carry any resistance genes or virulence genes.

***In vitro* and *in vivo* virulence of CR-hvKP isolates.** In the presence of pooled human serum, 11 CR-hvKP isolates showed different survival rates. Three isolates (RJ9582, RJ9717, RJ9299) exhibited grade 6, while RJ9337 exhibited a grade 3 response to serum killing. In contrast, the other seven isolates (RJ9950, RJ9860, RJ10129, RJ9490, RJ9690, RJ9752, RJ9846) showed grade 2 to serum killing (Fig. 5).

We infected *Galleria mellonella* larvae with an inoculum of $1 \times 10^5$ CFU for all the CR-hvKP outbreak isolates, as well as ATCC 25922 (negative-control strain of low virulence) and hvKP4 (positive-control isolate of high virulence). We observed that the survival of *G. mellonella* infected by two isolates (RJ9337 and RJ10129) was >50%, while survival of those infected by the other nine isolates was <40% at 72 h after infection, suggesting that these nine isolates were more virulent than RJ9337 and RJ10129 (Fig. 6A). Furthermore, the 50% lethal dose ($LD_{50}$) of these *K. pneumoniae* isolates were determined to vary from $10^4$ CFU to $10^6$ CFU, suggesting various levels of virulence. Certain clinical isolates (RJ9299, RJ9490, RJ9582, RJ9717, RJ9752, RJ9846, RJ9950) exhibited intermediate virulence, while the other

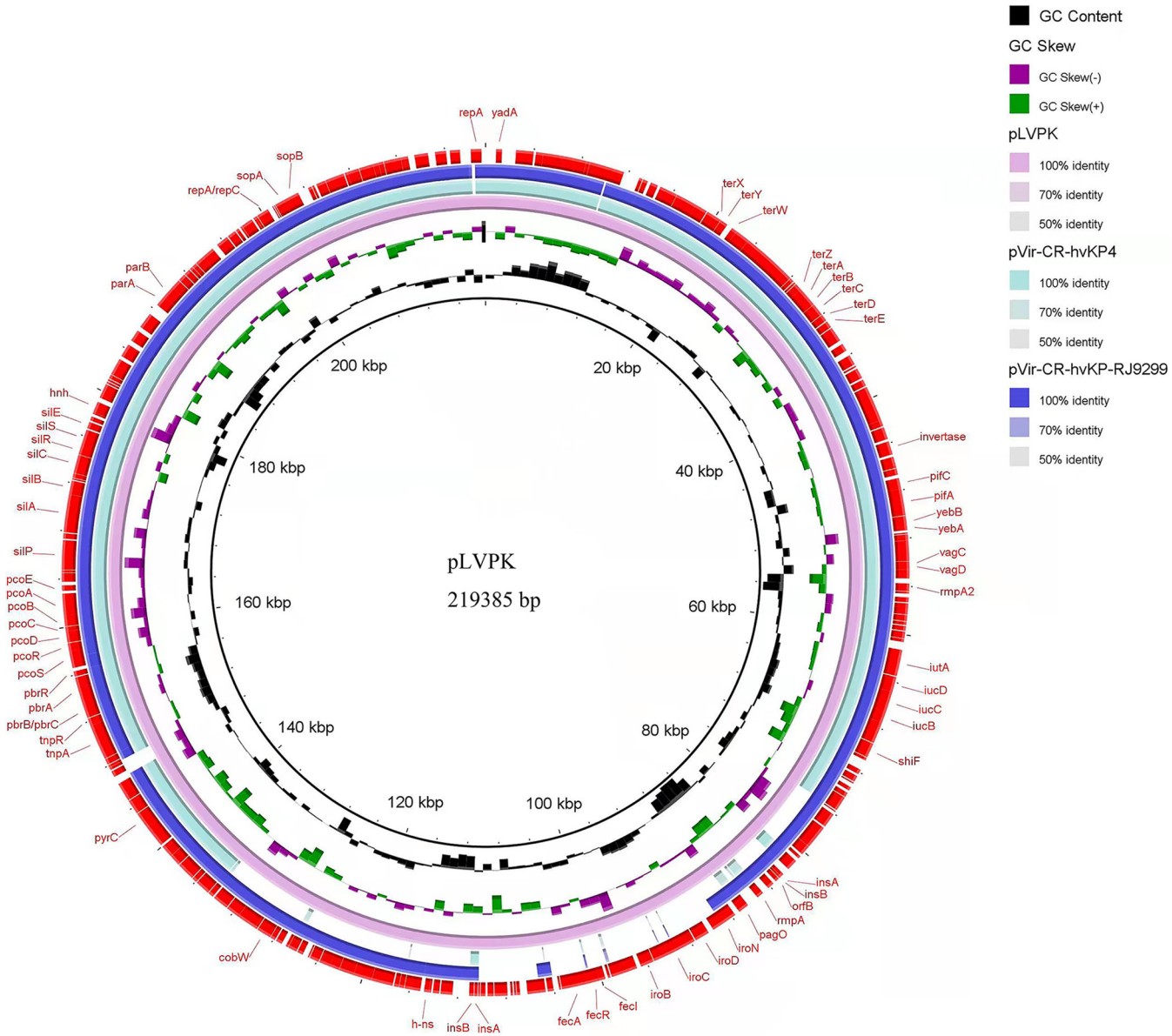

**FIG 3** Alignment of pVir-CR-hvKP-RJ9299 in the present study, plasmid pLVPK (GenBank accession no. AY378100), and plasmid pVir-CR-HvKP4 (GenBank accession no. MF437313) using BLAST Ring Image Generator (BRIG).

isolates (RJ9337, RJ9690, RJ9860, RJ10129) showed lower virulence with an LD$_{50}$ of $>10^6$ (Fig. 6B).

## DISCUSSION

Here, we report an outbreak caused by ST859 and K19, a novel ST clone, CR-hvKP isolates in a Chinese hospital. Since the first CR-hvKP was identified in Zhejiang in 2017, CR-hvKP have been increasingly reported in more countries besides China, including the United States, United Kingdom, Argentina, Japan, and Canada (13, 20–24). CR-hvKP were involved in a diverse range of STs and K-loci, including CRKP genetic background clones ST11 and K47/K64 and hvKP clones ST23/ST65/ST86 and K1/K2/K5, as well as other rare ST147, ST36, and ST25 types (21, 25). However, there was no literature on ST859 *K. pneumoniae* in public databases, let alone ST859 CR-hvKP. Hence, this is the first report about the microbiological features, virulence level, and genomic information of ST859 CR-hvKP isolates.

Based on the fact that ST859 was a single-locus variant of ST11, we suspect that ST859 might have evolved from a CRKP genetic background. In other words, the evolution

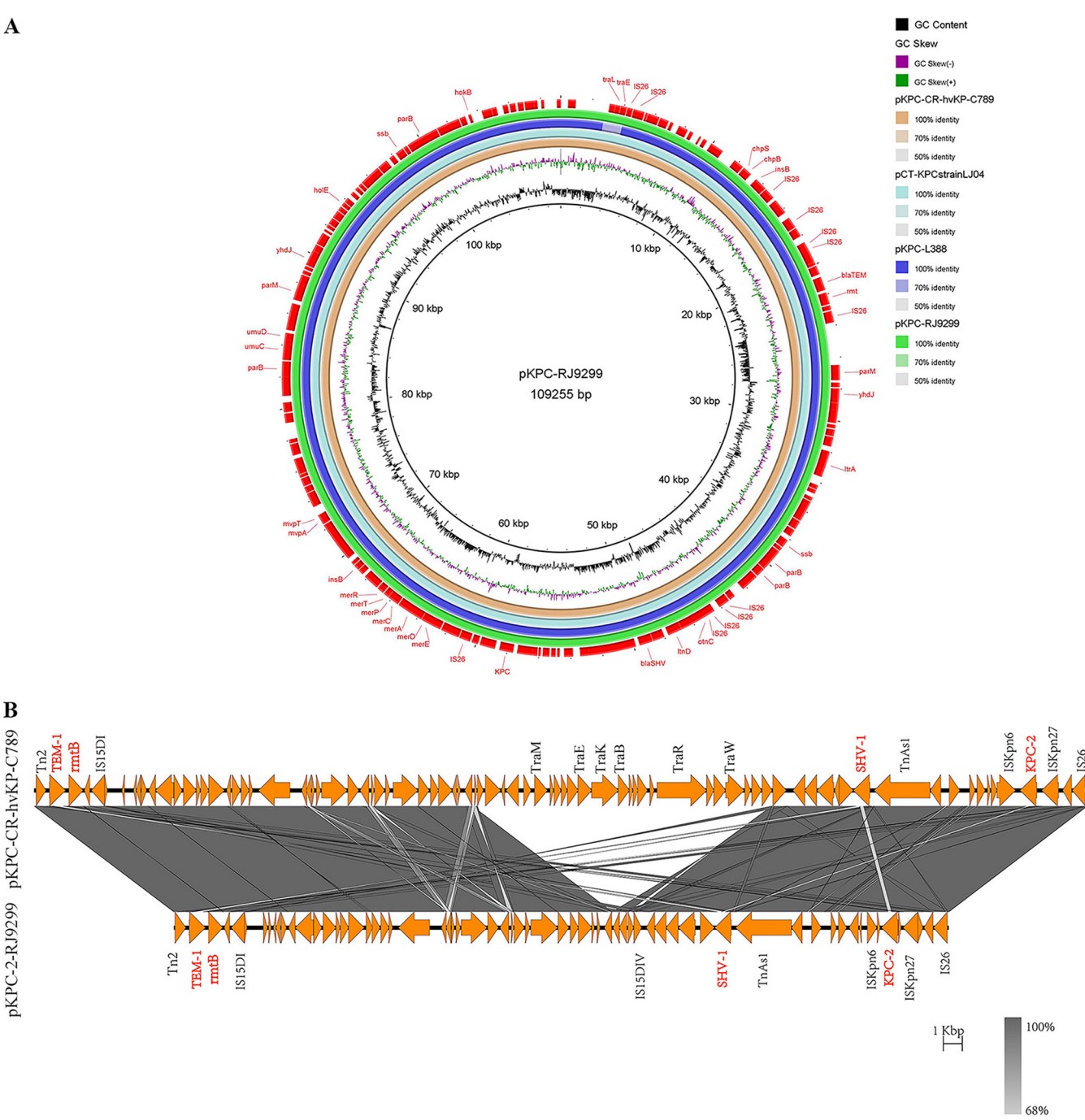

**FIG 4** Alignment of plasmids pKPC-L388 (accession version no. CP029225.1), pKPC-CR-hvKP-C789 (CP034417.1), and pKPC-2-RJ9299 recovered in the present study. (A) The circular map was generated using BRIG. (B) Colinear genome alignment of pKPC-CR-hvKP-C789 (CP034417.1) and pKPC-2-RJ9299. The program Easyfig was used in comparative genomics. Colored arrows indicate open reading frames (ORFs), and the shaded region reflects sequence similarity. The resistance genes are indicated in red.

process was probably that a KPC-2-producing ST859 CRKP strain acquired a virulence plasmid from hvKP and became CR-hvKP. As we know, nonconjugative virulence plasmids were usually confined to the classical hypervirulent *K. pneumoniae* (such as K1-ST23 or K2-CG43). However, they have been widely spotted in CRKP clones (such as ST11, ST15, and ST147) (26, 27). A recent study has demonstrated that the horizontal transfer of nonconjugative virulence plasmids could be mediated through the recombination between the virulence plasmid and the helper conjugative plasmid. In addition, the predicted oriT (origin of transfer) on a virulence plasmid could also be directly cleaved by the helper plasmid-encoded

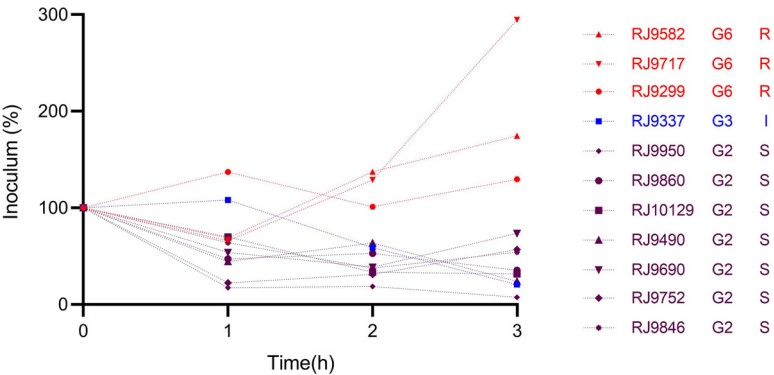

**FIG 5** Serum resistance results of 11 CR-hvKP isolates.

relaxase, and then the virulence plasmid is mobilized through the T4SS (type 4 secretion system) of the helper conjugative plasmid (28, 29). Since the plasmid backbones of the nonconjugative virulence plasmids from hypervirulent and resistant clones are quite conserved, they should share similar mechanisms for horizontal transfer. The current study showed that a new resistant clone, ST859, has also acquired the nonconjugative virulence plasmid and evolved into CR-hvKP. Generally, for this clone, the specific mechanisms of horizontal transfer of the virulence plasmid remain to be further investigated.

Instead of a typical 178-kb KPC-2-encoding plasmid, the size of pKPC-2-RJ9299 was 109 kb, which was a truncated version but has >90% similarity to a classic ~170-kb KPC-2-

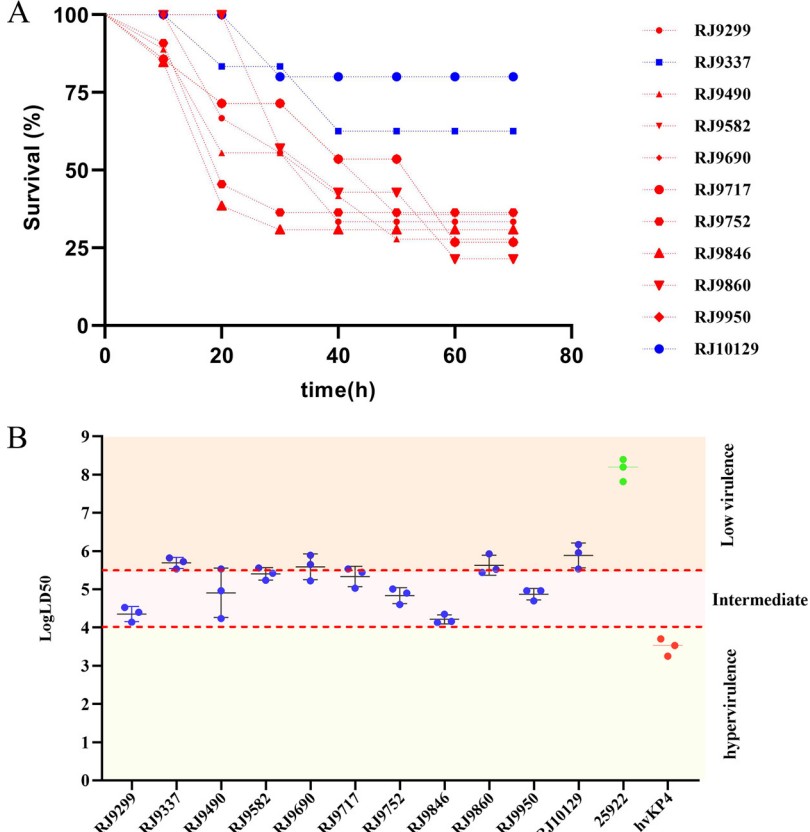

**FIG 6** Using the *G. mellonella* infection model to evaluate virulence of 11 CR-hvKP isolates. (A) Survival curves for *G. mellonella* infected with 11 CR-hvKP isolates. *G. mellonella* survival rate for those infected by two isolates (RJ9337 and RJ10129) was >50%, while survival for those infected by the other nine isolates was <40% at 72 h after infection. (B) Measurement of the $LD_{50}$ of clinical strains of *K. pneumoniae* in the *G. mellonella* infection model.

encoding plasmid. Reduction of the KPC-2-harboring plasmid length in CR-hvKP was inevitably accompanied by the loss of gene elements. Potentially, an ST859 CRKP underwent virulence-encoding plasmid acquisition, loss of certain genetic elements, and transposition, eventually contributing to the formation of CR-hvKP, which had better ability to adapt to the bacterial host. Also, genetic element loss during evolution is a performance of fitness cost. Whether ST859 will be a dominant prevalent clone for the dissemination of CR-hvKP isolates in the future remains unclear.

In contrast with Gu et al. (13), who reported a fatal outbreak caused by ST11 CR-hvKP which exhibited hypervirulence, our study showed that the $LD_{50}$s of seven CR-hvKP isolates varied from $10^4$ CFU to $10^6$ CFU (intermediate virulence); another four CR-hvKP isolates had lower virulence with $LD_{50}$s of $>10^6$ CFU. However, these isolates posed a great threat to the health of patients, and three patients died of CR-hvKP infection. According to the clinical history, patients RJ9690, RJ9752, and RJ9950 eventually died, although RJ9690 showed low virulence, with an $LD_{50}$ of $>10^6$, while RJ9752 and RJ9950 showed intermediate virulence, with $LD_{50}$s of $>10^{4-5.5}$ in *in vitro* virulence tests. Severe bloodstream infection might account for the death of patient RJ9690. Apart from infections caused by CR-hvKP isolates, severe underlying diseases might be another important cause for the death of the three patients. Morbidity and mortality rates for CRKP infections, especially ICU patients, are much higher than those for patients infected with classic *K. pneumoniae* (cKP), let alone the simultaneously hypervirulent, multidrug-resistant, and highly transmissible CR-hvKP isolates.

Given limited therapeutic drugs for clinical infection, effective infection prevention and control for these superbugs appeared to be particularly important. Current measures for infection prevention and control include hand hygiene, isolation of patients, environmental cleansing and equipment disinfection, and cautious use of antibiotics. Furthermore, we routinely screen carbapenem-resistant Enterobacteriaceae (CRE) in anal swab samples of patients before their admission to all the inpatient wards. Subsequent epidemiological investigation revealed that there was been no CRKP or CR-hvKP outbreak before November 2021. To the best of our knowledge, this is the first report of an outbreak caused by ST859 K19 CR-hvKP isolates in China. This observation is important, as resistance genes and virulence genes encoding mobile genetic elements were transmitting into more ST types in *K. pneumoniae*. Large-scale and extensive surveillance needs to be done to deeply study the evolution of CR-hvKP.

## MATERIALS AND METHODS

**Outbreak investigation.** Initially, we aimed to retrospectively investigate the molecular characteristics of *K. pneumoniae* isolated from Renji hospital. 120 CRKP isolates were collected between 1 January 2019 and 31 August 2020. String test was phenotypically used to screen hvKP, which was finally defined by aerobactin production. CR-hvKP was defined by both aerobactin production and resistance to any of the carbapenems. Of the 120 CRKP isolates, 11 were defined as CR-hvKP. Unexpectedly, these 11 CR-hvKP isolates were continuously isolated during 4 months. Furthermore, clinical data investigation showed that eight CR-hvKP isolates were isolated from the ICU, two patients were from neurosurgery, and one patient was from general surgery, which indicated this was probably an outbreak.

**Isolate identification and antimicrobial susceptibility testing.** Bacterial identification was conducted routinely using a Vitek-2 compact system by the clinical laboratory when the isolates were first isolated. We confirmed isolate identity via matrix-assisted laser desorption ionization–time of flight mass spectrometry (MALDI-TOF MS).

Antimicrobial susceptibility testing comprising 12 antibiotics was conducted for 11 CR-hvKP isolates using broth microdilution methodology. The breakpoint MIC of tigecycline was determined following the guidelines of the Clinical and Laboratory Standards Institute (CLSI-M100). ATCC 25922 was used as a quality control.

**Multilocus sequence typing.** Genetic diversity testing of 11 CR-hvKP isolates was performed with multilocus sequence typing (MLST). Seven housekeeping genes were amplified by PCR as described by the MLST database for *K. pneumoniae*. PCR products were sequenced and compared with that on the Pasteur Institute MLST website (https://bigsdb.pasteur.fr/cgi-bin/bigsdb/bigsdb.pl?db=pubmlst_klebsiella_seqdef&page=sequenceQuery).

**Whole-genome sequencing, phylogenetic construction, and drug resistance gene and virulence-associated gene analysis.** Chromosomal DNA of 11 CR-hvKP isolates was extracted using the TIANprep midi plasmid kit and was used for whole-genome sequencing (WGS). WGS was performed using the HiSeq X Ten PE150 sequencer platform (Illumina, USA) with a $2 \times 150$-bp read length (Majorbio Bio-Pharm Technology, Shanghai, China) for the 11 CR-hvKP isolates. Furthermore, RJ9299 were also long-read sequenced using PacBio RS II single molecule real-time (SMRT) technology. We trimmed and filtered the raw data before

assembly. Furthermore, adapter sequences, sequences containing more than 10% ambiguous N bases, and sequences shorter than 25 bp in length were also removed.

The phylogenetic tree was generated for 11 ST859 CR-hvKP isolates and 15 ST11 *K. pneumoniae* isolates (9 isolates from the First People's Hospital affiliated with Soochow University in 2020 and 6 isolates from Renji Hospital in 2020). Whole-genome alignment, SNP calling, and phylogenetic analysis were all done in CLC Genomics Workbench 12.0 (Qiagen) with the default options. The fully assembled sequence of RJ9299 (the isolate in this study) was used as the reference template for read mapping. A maximum likelihood (ML) tree using the general time reversible (GTR) model of nucleotide substitution with among-site rate heterogeneity across 4 categories (GTR+$\Gamma$) was constructed in CLC Genomics Workbench 12.0, followed by annotation using iTOL (https://itol.embl.de/) (30).

*De novo* assembly of the clean data was performed in CLC Genomics Workbench 12.0 (Qiagen) using the default options. The assembled contigs were employed in BLAST at the drug-resistant gene database (ResFinder database, 24 February 2021) and at the virulence factor database (VFDB, May 2021) in order to confirm if the resistance genes or virulence factors were present.

**Serum resistance assay.** Serum bactericidal activity was measured as previously described with minor modifications (31). First, 0.1 mL early-log-phase cells, washed and suspended in 0.9% NaCl (ca. $1 \times 10^6$ CFU/mL) bacteria were added to 0.1 mL pooled human serum from healthy volunteers. Viable counts were obtained at 0 h, 1 h, 2 h, and 3 h of incubation at 37°C on Mueller-Hinton agar. Each strain was tested at least three times.

***Galleria mellonella* infection.** The infection model of wax moth larvae ($\sim$300 mg *Galleria mellonella*) was used to test the virulence of 11 CR-hvKP isolates. *G. mellonella* larvae were purchased from Tianjin Huiyude Biotech Company (Tianjin, China) and divided into series of experimental groups (10/group). Log-phase cultures were washed and adjusted to concentrations of $1 \times 10^4$ CFU/mL, $1 \times 10^5$ CFU/mL, $1 \times 10^6$ CFU/mL, and $1 \times 10^7$ CFU/mL with phosphate-buffered saline (PBS). *G. mellonella* were infected with bacteria as described previously and the survival rate at 72 h postinfection was recorded. All experiments were conducted in triplicate.

**Data availability.** The complete whole-genome sequences of 11 CR-hvKP isolates have been deposited in the GenBank database under BioProject accession no. PRJNA799444.

## ACKNOWLEDGMENTS

This work was supported by the National Natural Science Foundation of China (grant no. 81702062) and the Scientific Research Project of Shanghai Municipal Health Commission (grant no. 20204Y0288).

We declare that there are no conflicts of interest.

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
