## [Reviewer comments · mSystems]

An outbreak of ST859-K19 Carbapenem-Resistant Hypervirulent *Klebsiella pneumoniae* in a Chinese teaching hospital

Junying Zhu, Xuemei Jiang, Lina Zhao, and Min Li

Corresponding Author(s): Min Li, Renji Hospital, Shanghai Jiaotong University School of Medicine

Review Timeline:

Submission Date:	October 26, 2021
Editorial Decision:	January 13, 2022
Revision Received:	February 1, 2022
Editorial Decision:	March 7, 2022
Revision Received:	March 15, 2022
Accepted:	April 20, 2022

Editor: Xiaoxia Lin

Reviewer(s): Disclosure of reviewer identity is with reference to reviewer comments included in decision letter(s). The following individuals involved in review of your submission have agreed to reveal their identity: Fuping HU (Reviewer #3)

Transaction Report:

DOI: <https://doi.org/10.1128/mSystems.01297-21>

January 13, 2022

Dr. Junying Zhu

Department of Laboratory Medicine, Ren Ji Hospital, School of Medicine, Shanghai Jiao Tong University, Shanghai 200127, China
Shanghai
China

Re: mSystems01297-21 (An outbreak of ST859-K19 **Carbapenem-Resistant Hypervirulent *Klebsiella pneumoniae* in a Chinese** teaching hospital)

Dear Dr. Junying Zhu:

Thank you for submitting your manuscript to mSystems. We have completed our review and will consider acceptance of your manuscript if you adequately address the reviewer comments, which are detailed below.

Preparing Revision Guidelines

Sincerely,

Xiaoxia "Nina" Lin

Editor, mSystems

Journals Department
Reviewer comments:

Reviewer #1 (Comments for the Author):

In this study, the authors collected 11 ST859 *Klebsiella pneumoniae* isolates, and considered it as CR-hvKP. It is a novel ST of CR-hvKP, and closely related to ST11. And it is the first report to depict the molecular and genomic characteristics of ST859 CR-hvKP isolates. The comments on this research are as follows:

Major comments:

1. In this study, so many isolates were collected from one patient, for example, RJ9229 had another 17 isolates from sputum, and only the first isolate was considered as CR-hvKP. How about the following isolates, and so as the other patients?
2. Lots of isolates were isolated from sputum, and the underlying diseases of patients included hypertension, cerebral hemorrhage and tumor. Since hvKP usually caused liver abscess, endophthalmitis, meningitis, and metastatic infectious diseases. How about your definition of hvKP, please discuss in the manuscript?
3. The analysis of Illumina and nanopore sequencing was different in this study, for example, in nanopore sequencing analysis, fimbriae, enterobactin, and yersiniabactin were considered as virulence factors, but not in Illumina sequencing analysis.
4. In figure 3, the reference was pVir-CR-hvKP-RJ9299, and the pink ring represented pLVPK, the cyan ring represented pVir-CR-hvKP4, and the blue ring represented the pVir-CR-hvKP-RJ9299. However, the coverage of pVir-CR-hvKP-RJ9299 was not 100%. The analysis had a mistake, please correct it.
5. Only 15 SNPs at most existed between these isolates, but the virulence level was different. It will be innovative to find the reasons determining the diversity of virulence.

Minor comments:

1. The authors identified the CR-hvKP with positive string test, so the first sentence "All eleven CR-hvKP outbreak isolates had a hypermucoviscous phenotype with positive string test" is not necessary.
2. Figure 1: I cannot find "another two isolates were respectively from blood and drainage" in the figure 1. Please correct it.
3. Figure 2: "The eleven ST859-K19 CR-hvKP isolates were clustered into clade I. Six ST11-K64 KPC-2 producing isolates which had positive string test, namely CR-hvKP, were clustered into clade III. Nine ST11-K64 KPC-2 producing isolates which had negative string test, namely CRKP, were clustered into clade II." I think the figure 2A had a mistake, and please clarify it.
4. Compared to pKPC-CR-hvKP-C789, RJ9299 had a deletion, please annotate the deletion region.

Reviewer #2 (Comments for the Author):

This is a description of an outbreak of *K. pneumoniae* ST859 (which is a single locus variant (SLV) of ST11) carrying a virulence plasmid and blaKPC-2, the latter in a separate plasmid to the former, among 11 patients, several of whom unfortunately died. For me, I find the insistence on this being novel because it is a new ST (it is an SLV of ST11, with probably only a single nucleotide change in the tonB gene) is overplayed and I would always exercise caution in describing an account as the first description. The other thing that really strikes me about this is that the virulence plasmid is so like pLVPK, which is a non-conjugative virulence plasmid associated with hypervirulent K2-CG43. So how did it get into this type? Usually, where you find virulence plasmids in 'high-risk' clones such as ST11, ST15, ST147 etc., they are hybrid virulence/resistance plasmids that have become conjugative through recombination (see for example Xie et al., 2020 A hybrid plasmid formed by recombination of a virulence plasmid and a resistance plasmid in *Klebsiella pneumoniae*. *J Glob Antimicrob Resist* 23:466-470), which is actually a more frightening prospect - with a non-conjugative plasmid the virulence plasmid is at least confined to that type. Maybe you could discuss this?

I worry about your defining isolates as hypervirulent on the basis of the string test - it is at best just an indication. There is no mention of how you did the nanopore sequencing; also did you use the fully assembled sequence of RJ9299 as reference, or just the Illumina sequence? Is the sequence of RJ9299 on GenBank (or similar) - if not, it needs to be and an accession number provided.

Other specific points

'CR-hvKP has been reported in an increasing number of STs, including ST23, ST65, ST1797, ST43, ST231 and ST307' There are quite a few more - like ST147, ST15, ST383... The following review paper may be helpful to include Yang X, Dong N, Chan EW, Zhang R, Chen S. Carbapenem Resistance-Encoding and Virulence-Encoding Conjugative Plasmids in *Klebsiella pneumoniae*. *Trends Microbiol*. 2021 Jan;29(1):65-83.

'while ST258 was the main prevalent clone in North America, Latin America, and several European countries (6-9). ' Can't help but think that this is outdated and overplayed - it certainly is not my experience in the UK.

'All the patients

received antimicrobial treatment, including carbapenem alone or in the combination with other alternative antibiotics when necessary' Why were patients treated with carbapenem alone for a carbapenemase producing organism?

fosA is intrinsic in *K. pneumoniae*

'In terms of

virulence genes, they all carried aerobactin (*iucABCDiutA*) and salmochelin (*iroE*) regulator of mucoid phenotype A (*rmpA* and *rmpA2*)' - but the virulence plasmid carried *iroBCDN*; surely that should be

mentioned here. Not sure about this chromosomal *iroE*.
Number of SNPs - please say how many bases were compared.

Reviewer #3 (Comments for the Author):

This study is the first to report the prevalence and dissemination of ST859 CR-hvKp strains. The characteristics of 11 CR-hvKp strains were analyzed by DNA sequencing, and genome-wide analysis was performed, which has important reference value for clinical anti-infection treatment and nosocomial infection control. I have some comments as follows.

1. The MIC data in Table 1 is the exact MIC value, or has the prefix " \leq " or " \geq "? which is inconsistent with the description of the data in the text. In the text, it shows " ≥ 64 ", " ≥ 128 " and " ≤ 0.25 " etc. For example, not all 11 strains of imipenem have a MIC of " ≥ 64 ", and the MIC range should be 32- ≥ 128 mg/L according to table 1.
2. Table 2 shows 12 strains of CR-hvKP, but in the text and Table 1 show 11 strains of CR-hvKP, with one more strain of RJ 10091. Resistance genes should also describe the presence or absence of plasmid-mediated quinolone and aminoglycoside resistance genes.
3. This study needs to supplement the clinical history of 11 patients, including patient demographic data, isolation date, specimen source, antibiotic treatment, underlying diseases, and outcomes. The medical history data can also be preliminarily compared with the virulence test results to observe whether strains with strong in vitro virulence are more likely to cause the death of patients.

Reviewer #1

1. In this study, so many isolates were collected from one patient, for example, RJ9229 had another 17 isolates from sputum, and only the first isolate was considered as CR-hvKP. How about the following isolates, and so as the other patients?

Among the eleven patients, there were three patients from whom we also collected another CR-hvKP isolate besides CR-hvKP in our study. RJ9950 had a ST859 CR-hvKP after RJ9950 isolate, RJ10129 had a ST412 CR-hvKP isolate before RJ10129 isolate. RJ9690 had a ST218 CR-hvKP isolate and a ST 859 CR-hvKP isolate before and after RJ9690. These information has added to Fig1. The other eight patients had no another CR-hvKP isolates before and after CR-hvKP in our study.

2. Lots of isolates were isolated from sputum, and the underlying diseases of patients included hypertension, cerebral hemorrhage and tumor. Since hvKP usually caused liver abscess, endophthalmitis, meningitis, and metastatic infectious diseases. How about your definition of hvKP, please discuss in the manuscript?

Until today, there is no unified and standard definition of hvKP. In the present study, string test was phenotypically used to screen hvKP, which was finally defined by aerobactin production. CR-hvKP was defined by both aerobactin production and resistance to any of the carbapenems. Please see the MATERIALS AND METHODS section which marked in blue in “Marked-Up Manuscript”.

3. The analysis of Illumina and nanopore sequencing was different in this study, for example, in nanopore sequencing analysis, fimbriae, enterobactin, and yersiniabactin were considered as virulence factors, but not in Illumina sequencing analysis.

Sorry, we used PacBio platform, not nanopore for RJ9299 sequencing. We have corrected it. In fact, the analysis of Illumina and PacBio sequencing was same. I only wrote some important virulence factors in the eleven isolates sequenced by Illumina platform. Now, I have added all the virulence factors in the “Marked-Up Manuscript”. Please see the RESULTS section (Resistance genes, virulence genes profiles and phylogenetic analysis) which marked in blue in “Marked-Up Manuscript”.

4. In figure 3, the reference was pVir-CR-hvKP-RJ9299, and the pink ring represented pLVPK, the cyan ring represented pVir-CR-hvKP4, and the blue ring represented the pVir-CR-hvKP-RJ9299. However, the coverage of pVir-CR-hvKP-RJ9299 was not 100%. The analysis had a mistake, please correct it.

Sorry, it is a mistake. The reference was pLVPK. Please see the figure 3. We have corrected it.

5. Only 15 SNPs at most existed between these isolates, but the virulence level was different. It will be innovative to find the reasons determining the diversity of virulence.

Yes, it is interesting that 15 SNPs at most existed between these isolates, but the virulence level was different. We will investigate the mechanism of virulence difference between these isolates.

Minor comments:

1. The authors identified the CR-hvKP with positive string test, so the first sentence "All eleven CR-hvKP outbreak isolates had a hypermucoviscous phenotype with positive string test" is not necessary.

Thank you for your close examination. We have corrected it. Please see the RESULTS section in "Marked-Up Manuscript".

2. Figure 1: I cannot find "another two isolates were respectively from blood and drainage" in the figure 1. Please correct it.

RJ9337 was from drainage and RJ9690 was from blood. We have corrected it in the Figure 1. Furthermore, We have supplemented the clinical history of eleven patients in Table 1. There is also specimen source of the eleven isolates in Table 1.

3. Figure 2: "The eleven ST859-K19 CR-hvKP isolates were clustered into clade I. Six ST11-K64 KPC-2 producing isolates which had positive string test, namely CR-hvKP, were clustered into clade III. Nine ST11-K64 KPC-2 producing isolates which had negative string test, namely CRKP, were clustered into clade II." I think the figure 2A had a mistake, and please clarify it.

Thank you for your close examination. We have corrected it. The right is "ST11-K64 KPC-2 producing isolates which had positive string test, namely CR-hvKP, were clustered into clade II. Nine ST11-K64 KPC-2 producing isolates which had negative string test, namely CRKP, were clustered into clade III." Please see the RESULTS section which marked in blue in "Marked-Up Manuscript".

4. Compared to pKPC-CR-hvKP-C789, RJ9299 had a deletion, please annotate the deletion region.

We have annotate the deletion region in figure 4B.

Reviewer #2

1. This is a description of an outbreak of *K. pneumoniae* ST859 (which is a single locus variant (SLV) of ST11) carrying a virulence plasmid and blaKPC-2, the latter in a separate plasmid to the former, among 11 patients, several of whom unfortunately died. For me, I find the insistence on this being novel because it is a new ST (it is an SLV of ST11, with probably only a single nucleotide change in the tonB gene) is overplayed and I would always exercise caution in describing an account as the first description.

Yes, it should be cautious for us to use the first time to describe ST859 (an SLV of ST11) isolates. We use the first time because we blasted *tonB* gene in ST859 and *tonB* gene in ST11 and found there were many nucleotide changes, but not a single nucleotide change, between the two *tonB* genes. The image below (left) showed nucleotide blast results of *tonB* genes between RJ9860 isolate (in our study) and a ST11 isolate. Furthermore, we have searched PubMed by keywords “*Klebsiella pneumoniae* and ST859”, there was no literature published in PubMed. The image below (right) showed the PubMed search result. So I described ST859 isolates as the first description

Score	Expect	Identities	Gaps	Strand
937	0.0	528/538(98%)	2/538(0%)	
Query 1	6	GTCTCAGGTTATTCACAGCTTCTCTGAGCGCCGATAGATCAATGATGGCC	62	
Subject 8	6	GTCTCAGGTTATTCACAGCTTCTCTGAGCGCCGATAGATCAATGATGGCC	62	
Query 63	63	GGGATCTTAGCGCCCTGGGGGCGCCCTCTGAGCCCTTGTAAAGCGA	122	
Subject 66	66	GGGATCTTAGCGCCCTGGGGGCGCCCTCTGAGCCCTTGTAAAGCGA	122	
Query 123	123	ACTTACGCGAGCCGAGGATCTGCTGACCCGCAAGAGCGCGGTGATGCA	182	
Subject 126	126	ACTTACGCGAGCCGAGGATCTGCTGACCCGCAAGAGCGCGGTGATGCA	182	
Query 183	183	TAAACCGAAGCTTAGTAAAGCCAACTTAAGCTTAACTGAGGAAAGCT	242	
Subject 186	186	TAAACCGAAGCTTAGTAAAGCCAACTTAAGCTTAACTGAGGAAAGCT	242	
Query 243	243	TGACAGCGAGCGGAGCTGAGCGGCGAGCGCCGCTGCGCTTGA	302	
Subject 246	246	TGACAGCGAGCGGAGCTGAGCGGCGAGCGCCGCTGCGCTTGA	302	
Query 303	303	AAGAGATATGGGGGCGCGCGCGAGCGAGGACTGACGAGGCTAAAGT	362	
Subject 306	306	AAGAGATATGGGGGCGCGCGCGAGCGAGGACTGACGAGGCTAAAGT	362	
Query 363	363	CAGCTTCTCTCAGCGCCGAGGATGAGCGGCTGAGCGCTATGAG	422	
Subject 366	366	CAGCTTCTCTCAGCGCCGAGGATGAGCGGCTGAGCGCTATGAG	422	
Query 423	423	GGGCTGAGCGGCTGCTTAAAGCTGGTACGAGTAACTTCACTTGGCTGA	482	
Subject 426	426	GGGCTGAGCGGCTGCTTAAAGCTGGTACGAGTAACTTCACTTGGCTGA	482	
Query 483	483	TGGGCTATGATATGCGAGATCTCTCTCAGCGGCGAAGAAATCTTAA	542	
Subject 486	486	TGGGCTATGATATGCGAGATCTCTCTCAGCGGCGAAGAAATCTTAA	542	

left: results of *tonB* genes between RJ9950 isolate and a ST11 isolate

Right: PubMed search result by keywords “*Klebsiella pneumoniae* and ST859”

2. The other thing that really strikes me about this is that the virulence plasmid is so like pLVPK, which is a non-conjugative virulence plasmid associated with hypervirulent K2-CG43. So how did it get into this type? Usually, where you find virulence plasmids in 'high-risk' clones such as ST11, ST15, ST147 etc., they are hybrid virulence/resistance plasmids that have become conjugative through recombination (see for example Xie et al., 2020 A hybrid plasmid formed by recombination of a virulence plasmid and a resistance plasmid in *Klebsiella pneumoniae*. J Glob Antimicrob Resist 23:466-470), which is actually a more frightening prospect - with a non-conjugative plasmid the virulence plasmid is at least confined to that type. Maybe you could discuss this?

Yes, it is a very good advice for us to discuss how the non-conjugative virulence plasmid get into ST859 isolates. Regarding the question how a non-conjugative virulence plasmid was not confined to 'high-risk' clones such as ST11, ST15, ST147 etc, but get into ST859 clone, it is probably the non-conjugative virulence plasmid in ST859 isolates was mobilized by the conjugative plasmid, such as incompatibility group F (IncF), from the hvKP strain into ST859 CRKP strains or by employing intermediate *E. coli* strains according to Xu et al., 2021 Mobilization of the nonconjugative virulence plasmid from hypervirulent *Klebsiella pneumoniae*. However, this procession should be further validated. We have added this content in the Discussion section marked in purple in “Marked-Up Manuscript”.

3. I worry about your defining isolates as hypervirulent on the basis of the string test - it is at best just an indication.

Yes, as a phenotypic detection method, string test is just an indication for hypervirulent isolates. In our study, hvKP was defined by aerobactin detection according to Zhang Y et al., 2016 High

Prevalence of Hypervirulent *Klebsiella pneumoniae* Infection in China: Geographic Distribution, Clinical Characteristics, and Antimicrobial Resistance. *Antimicrob Agents Chemother.* 60:6115-20. doi: 10.1128/AAC.01127-16.

Please see the MATERIALS AND METHODS section which marked in purple in “Marked-Up Manuscript”.

4. There is no mention of how you did the nanopore sequencing; also did you use the fully assembled sequence of RJ9299 as reference, or just the Illumina sequence? Is the sequence of RJ9299 on GenBank (or similar) - if not, it needs to be and an accession number provided.

Sorry, we used PacBio platform, not nanopore for RJ9299 sequencing. We have corrected it. This is our negligence for no mention of how we did the PacBio sequencing. We have added this content in the MATERIALS AND METHODS section marked in purple in “Marked-Up Manuscript”.

We used the fully assembled sequence of RJ9299 as reference.

The complete whole-genome sequences of eleven CR-hvKP isolates have been deposited in the GenBank database under BioProject accession no. PRJNA799444. Please see the Data availability section which marked in purple in “Marked-Up Manuscript”.

5. ST23, ST65, ST1797, ST43, ST231 and ST307' There are quite a few more - like ST147, ST15, ST383... The following review paper may be helpful to include Yang X, Dong N, Chan EW, Zhang R, Chen S. Carbapenem Resistance-Encoding and Virulence-Encoding Conjugative Plasmids in *Klebsiella pneumoniae*. *Trends Microbiol.* 2021 Jan;29(1):65-83.

Yes, it is a very good suggestion for us to refer to this review. We have supplemented ST147, ST15, ST383, ST268, ST595, ST375, ST48 in this section which marked in purple in “Marked-Up Manuscript”.

6. 'while ST258 was the main prevalent clone in North America, Latin America, and several European countries (6-9). ' Can't help but think that this is outdated and overplayed - it certainly is not my experience in the UK.

Yes, we have searched PubMed and found: ST11 clone is predominantly found in China and South America, ST258 is mostly reported in the United States, ST512 is endemic in Italy and Greece, ST147 is mainly reported in India and Tunisia. Please see this section which marked in purple in “Marked-Up Manuscript”.

7. All the patients received antimicrobial treatment, including carbapenem alone or in the combination with other alternative antibiotics when necessary' Why were patients treated with carbapenem alone for a carbapenemase producing organism?

In the case of severe infection, patients were empirically and prophylactically treated with carbapenem before the CRKP isolates were cultured. Furthermore, for infection caused by CRKP which showed low level resistance to carbapenem, carbapenem in combination with other drugs is

also a good therapy in the case of limited available antibiotics.

8. *fosA* is intrinsic in *K. pneumoniae*

Sorry, because of the limited knowledge about *fosA*, we do not have a clear understanding about *fosA* is intrinsic in *K. pneumoniae*. We will be very grateful if respectable reviewer can provide us a reference article.

9. 'In terms of virulence genes, they all carried aerobactin (*iucABCDiutA*) and salmochelin (*iroE*) regulator of mucoid phenotype A (*rmpA* and *rmpA2*)' - but the virulence plasmid carried *iroBCDN*; surely that should be mentioned here. Not sure about this chromosomal *iroE*.

Sorry, it is a mistake. All the eleven isolates sequenced by Illumina carried aerobactin (*iucABCDiutA*), salmochelin (*iroE*) and regulator of mucoid phenotype A (*rmpA* and *rmpA2*). For RJ9299 sequenced by Illumina and PacBio, the chromosome carried *iroE*, not *iroBCDN*. There were no *iroBCDN* genes in RJ9299. We have corrected it. Please see this section which marked in blue in "Marked-Up Manuscript".

10. Number of SNPs - please say how many bases were compared.

5901 core genome SNPs between the 11 ST859 outbreak isolates and 15 genomes of ST11 *K. pneumoniae* were identified and used to construct the maximum likelihood tree. We have added this content in RESULTS section which marked in purple in "Marked-Up Manuscript".

Reviewer #3:

This study is the first to report the prevalence and dissemination of ST859 CR-hvKp strains. The characteristics of 11 CR-hvKp strains were analyzed by DNA sequencing, and genome-wide analysis was performed, which has important reference value for clinical anti-infection treatment and nosocomial infection control. I have some comments as follows.

1. The MIC data in Table 1 is the exact MIC value, or has the prefix "{less than or equal to}" or "{greater than or equal to}"? which is inconsistent with the description of the data in the text. In the text, it shows "{greater than or equal to}64", "{greater than or equal to}128" and "{less than or equal to} 0.25" etc. For example, not all 11 strains of imipenem have a MIC of "{greater than or equal to}64", and the MIC range should be 32-"{greater than or equal to}128 mg/L according to table 1.

The MIC data in Table 1 is the exact MIC value. We have corrected this section according to the requirements of reviewer. Please see this section which marked in red in "Marked-Up Manuscript".

2. Table 2 shows 12 strains of CR-hvKp, but in the text and Table 1 show 11 strains of CR-hvKp, with one more strain of RJ 10091. Resistance genes should also describe the presence or absence of plasmid-mediated quinolone and aminoglycoside resistance genes.

Thank you for your close examination. We have deleted the extra strain of RJ 10091 in Table 2. We have described the presence of plasmid-mediated quinolone and aminoglycoside resistance genes in text and in Table 2. Please see this section which marked in red in “Marked-Up Manuscript”.

3. This study needs to supplement the clinical history of 11 patients, including patient demographic data, isolation date, specimen source, antibiotic treatment, underlying diseases, and outcomes. The medical history data can also be preliminarily compared with the virulence test results to observe whether strains with strong in vitro virulence are more likely to cause the death of patients.

Yes, we have supplemented the clinical history of 11 patients in Table 1 of the new version. Furthermore, we have compared the medical history data with the virulence test results to discuss whether strains with strong in vitro virulence are more likely to cause the death of patients. Please see DISCUSSION section which marked in red in “Marked-Up Manuscript”.

March 7, 2022

Dr. Min Li
Renji Hospital, Shanghai Jiaotong University School of Medicine
Department of Laboratory Medicine
shanghai
China

Re: mSystems01297-21R1 (An outbreak of ST859-K19 **Carbapenem-Resistant Hypervirulent *Klebsiella pneumoniae* in a Chinese** teaching hospital)

Dear Dr. Min Li:

Thank you for submitting your revised manuscript to mSystems. I have completed the review and discussed with a senior editor. We concluded that you and your co-authors have adequately addressed most of the previous reviewer comments. However, the reviewers have raised new issues (see details below). One major concern is that description of details for the methods is lacking. Please address these issues carefully in your next revision.

Below you will find instructions from the mSystems editorial office and comments generated during the review.

Preparing Revision Guidelines

Sincerely,

Xiaoxia "Nina" Lin

Editor, mSystems

Journals Department
Reviewer comments:

Reviewer #1 (Comments for the Author):

There are some limited issues in this study, procedures are described so superficially that it is not possible to assess them. There were several reasons as follows causing the phylogenetic results inconsistent with the virulence results, including the choice of reference genome and infection model, and the recombination of genome.

Reviewer #2 (Comments for the Author):

Thank you for making changes to your paper. However, the following is incorrect and MUST be changed:

'As we know, non-conjugative

virulence plasmids were usually confined to 'high-risk' clones such as ST11, ST15, ST147 etc, but how did they get into ST859 clone?'

They are ABSOLUTELY NOT IN high-risk clones (which when they carry virulence plasmids are conjugative, hybrid resistance/virulence plasmids and definitely not non-conjugative plasmids) but they are confined to the classical hypervirulent types (such as K2-CG43 or K1-ST23); hence the clear association between virulence and those hypervirulent types.

How can a 'non-conjugative virulence plasmid in ST859 isolates were mobilized by the conjugative plasmid' when the plasmid doesn't appear to be conjugative at all? That is the remarkable thing about your description - if the plasmid is non-conjugative there is no obvious explanation as to how it has moved to a new type (the ST859) and what you have written really doesn't make any sense.

Reviewer #1 (Comments for the Author):

There are some limited issues in this study, procedures are described so superficially that it is not possible to assess them. There were several reasons as follows causing the phylogenetic results inconsistent with the virulence results, including the choice of reference genome and infection model, and the recombination of genome.

Thank you for your kind advice. We have revised the details of methods and results to make it clear. Please see the section which was marked in red in “Marked-Up Manuscript”. Furthermore, I will explain some confusing issues below.

Why do we include ST11 *K. pneumoniae* isolates to generate phylogenetic tree?

As we know, ST859 was a single locus variant of ST11. However, no ST859 genome data was available in GenBank, and we have to choose the ST11 isolates as well as ST859 isolates in our study to generate phylogenetic tree.

Why do we choose the fully assembled sequence of RJ9299 as reference genome?

RJ9299 was isolated earlier and displayed higher virulence than other 10 CR-hvKP isolates in this study. As we mentioned before, no ST859 *K. pneumoniae* genome data was available in GenBank. Since the reference strain should have similar genetic backgrounds with those we want to determine, RJ9299 was then employed as the reference to determine the phylogenetic relationship with other ST859 CR-hvKP isolates. Therefore, RJ9299 was chosen as the reference genome.

Why do we choose *G. mellonella* infection model?

We certainly know that mice infection model was better when evaluating bacterial virulence. However, we have 11 isolates to do virulence experiment. It is impracticable and unreasonable to use so many mice. In contrast, *G. mellonella* has an advantage in accessibility and convenience for a virulence experiment. Furthermore, *G. mellonella* infection model has been widely used to evaluate bacterial virulence (For example Yang X, et al. A conjugative plasmid that augments virulence in *Klebsiella pneumoniae*. *Nat Microbiol.* 2019;4(12):2039-2043.). So we choose *G. mellonella* infection model to evaluate the virulence of isolates.

The problems of phylogenetic results inconsistent with the virulence results?

The phylogenetic tree was generated for 11 ST859 isolates in our study and 15 ST11 *K. pneumoniae* isolates (nine isolates from the First People's Hospital affiliated to Soochow University in 2020 and six isolates from Renji Hospital in 2020). The virulence of 11 ST859 isolates in our study have been evaluated using *G. mellonella* infection model. However, for these 15 ST11 isolates, only string test has been performed. So we linked the phylogenetic results with the STs, K-locus type, KPC-2 producing and string test of all the isolates (Fig 2A).

As Fig 2A showed, The population framework revealed a diverse population structure containing three distinct clades (clade I, clade II, clade III). The eleven ST859-K19 CR-hvKP isolates were clustered into clade I. Six ST11-K64 KPC-2 producing isolates which had positive string test, namely CR-hvKP, were clustered into clade II. Nine ST11-K64 KPC-2 producing isolates which had negative string test, namely CRKP, were clustered into clade III. Furthermore,

the three clades could be markedly distinguished in the unrooted tree (Fig. 2B).

In this respect, the phylogenetic results were consistent with the virulence phenotypes.

FIG 2 Phylogenetic structure and paired SNP distance of *K. pneumoniae* (A) Maximum likelihood tree of ST859 and ST11 clonotype *K. pneumoniae* isolates. (B) Unrooted tree of ST11 and ST859 clonotype *K. pneumoniae* isolates. Branches could be classified into three clades. (C) Paired SNP distance of ST859 clonotype *K. pneumoniae* isolates.

Reviewer #2 (Comments for the Author):

1. Thank you for making changes to your paper. However, the following is incorrect and MUST be changed: 'As we know, non-conjugative virulence plasmids were usually confined to 'high-risk' clones such as ST11, ST15, ST147 etc, but how did they get into ST859 clone?' They are ABSOLUTELY NOT IN high-risk clones (which when they carry virulence plasmids are conjugative, hybrid resistance/virulence plasmids and definitely not non-conjugative plasmids) but they are confined to the classical hypervirulent types (such as K2-CG43 or K1-ST23); hence the clear association between virulence and those hypervirulent types.

Thanks for the reviewer's comments. We have corrected it in the Discussion section which was marked in blue.

2. How can a 'non-conjugative virulence plasmid in ST859 isolates were mobilized by the conjugative plasmid' when the plasmid doesn't appear to be conjugative at all? That is the remarkable thing about your description - if the plasmid is non-conjugative there is no obvious explanation as to how it has moved to a new type (the ST859) and what you have written really doesn't make any sense.

As the reviewer mentioned, non-conjugative virulence plasmids were usually confined to the classical hypervirulent *K. pneumoniae* (such as K1-ST23 or K2-CG43). However, they have been

widely spotted in carbapenem resistant *K. pneumoniae* (such as ST11, ST15 or ST147). A recent study has demonstrated that the horizontal transfer of non-conjugative virulence plasmid could be mediated through the recombination between the virulence plasmid and the helper conjugative plasmid. Besides, the predicted oriT (origin of transfer) on virulence plasmid could also be directly cleaved by the helper plasmid-encoded relaxase, and then the virulence plasmid is mobilized through the T4SS (type 4 secretion system) of the helper conjugative plasmid. The current study just displayed that a new resistant clone, ST859, has also acquired the non-conjugative virulence plasmid and evolved into carbapenem resistant hypervirulent *K. pneumoniae* (CR-hvKP). Generally, for this clone, the specific mechanisms of horizontal transfer of the virulence plasmid remain to be further investigated. This part has also been added to the Discussion section which was marked in blue.

April 20, 2022

Dr. Min Li
Renji Hospital, Shanghai Jiaotong University School of Medicine
Department of Laboratory Medicine
shanghai
China

Re: mSystems01297-21R2 (An outbreak of ST859-K19 **Carbapenem-Resistant Hypervirulent *Klebsiella pneumoniae* in a Chinese** teaching hospital)

Dear Dr. Min Li:

I am delighted to inform you that your manuscript has been accepted, and I am forwarding it to the ASM Journals Department for publication. For your reference, ASM Journals' address is given below. Before it can be scheduled for publication, your manuscript will be checked by the mSystems production staff to make sure that all elements meet the technical requirements for publication. They will contact you if anything needs to be revised before copyediting and production can begin. Otherwise, you will be notified when your proofs are ready to be viewed.

Publication Fees:

We recognize that the video files can become quite large, and so to avoid quality loss ASM suggests sending the video file via <https://www.wetransfer.com/>. When you have a final version of the video and the still ready to share, please send it to mSystems staff at mssystems@asmusa.org.

For mSystems research articles, if you would like to submit an image for consideration as the Featured Image for an issue, please contact mSystems staff at mssystems@asmusa.org.

Sincerely,

Xiaoxia "Nina" Lin

Editor, mSystems

Journals Department
Phone: (202) 737-3600